# Temperature-related death burden of various neurodegenerative diseases under climate warming: a nationwide modelling study

Peng Yin[1,5], Ya Gao[2,5], Renjie Chen [2,5], Wei Liu[1], Cheng He [2], Junwei Hao [3,4] ✉, Maigeng Zhou [1] ✉ & Haidong Kan [2] ✉

Limited knowledge exists regarding the ramifications of climate warming on death burden from neurodegenerative diseases. Here, we conducted a nationwide, individual-level, case-crossover study between 2013 and 2019 to investigate the effects of non-optimal temperatures on various neurodegenerative diseases and to predict the potential death burden under different climate change scenarios. Our findings reveal that both low and high temperatures are linked to increased risks of neurodegenerative diseases death. We project that heat-related neurodegenerative disease deaths would increase, while cold-related deaths would decrease. This is characterized by a steeper slope in the high-emission scenario, but a less pronounced trend in the scenarios involving mitigation strategies. Furthermore, we predict that the net changes in attributable death would increase after the mid-21st century, especially under the unrestricted-emission scenario. These results highlight the urgent need for effective climate and public health policies to address the growing challenges of neurodegenerative diseases associated with global warming.

Climate change is recognized as the greatest challenge of the twenty-first century, threatening human health and socioeconomic development[1]. As a direct consequence of climate change, overheating has a wide range of health effects, often exacerbating pre-existing diseases or pathophysiological conditions and leading to premature deaths and disability[2,3]. A growing body of studies have suggested that the future burden of heat-related mortality or the net burden due to non-optimum temperature would increase significantly under a warming climate[4–6]. Neurodegenerative diseases are a broad category of neurological disorders characterized by the progressive

degeneration of the structure and function of the central or peripheral nervous system, which most notably include Alzheimer's disease (AD), non-Alzheimer's dementias, and Parkinson's disease (PD)[7]. Given that neurodegenerative diseases constitute the major cause of disability-adjusted life-years lost[8], exploring the impact of non-optimal temperatures on neurodegenerative diseases under climate warming has important public health implications, especially in an era of ageing.

A few previous epidemiological studies have associated high or (and) low ambient temperature with increased hospitalization and mortality from the overall neurodegenerative diseases or a specific

[1]National Center for Chronic Noncommunicable Disease Control and Prevention, Chinese Center for Disease Control and Prevention, Beijing, China. [2]School of Public Health, Key Lab of Public Health Safety of the Ministry of Education, NHC Key Lab of Health Technology Assessment, IRDR ICoE on Risk Inter-connectivity and Governance on Weather/Climate Extremes Impact and Public Health, Fudan University, Shanghai, China. [3]Department of Neurology, Xuanwu Hospital, Capital Medical University, Beijing, China. [4]National Center for Neurological Disorders, Beijing, China. [5]These authors contributed equally: Peng Yin, Ya Gao, Renjie Chen. ✉e-mail: haojunwei@vip.163.com; zhoumaigeng@ncncd.chinacdc.cn; kanh@fudan.edu.cn

disease[9–15]. However, the studied population is relatively limited and the findings are inconsistent and even contradictory[16–19]. Furthermore, all the existing evidence was derived from ecological time-series studies based on daily aggregate health data, resulting in apparent ecological fallacy and residual bias by individual-level confounders. More importantly, it remains unclear on how future burden of neurodegenerative diseases death associated with non-optimum temperature would vary in a changing climate. Only one study from England estimated that heat-related dementia hospital admissions would nearly triple by 2040 compared to the baseline period under a high-emission scenario[20]. Consequently, these gaps make it difficult to fully understand the death risks of neurodegenerative diseases associated with non-optimum temperature, and to comprehensively capture the overall impact of future warming on the death burden of neurodegenerative diseases. Such evidence is critically important to the development and coordination of evidence-based public health policies to better address the complicated problems induced by global climate change and population ageing.

In this work, by virtue of the China's nationwide death registry, we conducted a large-scale, individual-level, case-crossover study to explore the associations between non-optimal temperatures and death from various neurodegenerative diseases. Additionally, we predicted the death burden in different climate zones under multiple climate change scenarios. Our findings indicate that both high and low temperatures are related with increased death risks of multiple neurodegenerative diseases. Furthermore, we demonstrate that climate warming will lead to a substantial increase in the heat-related deaths from neurodegenerative diseases, while cold-related deaths will decrease. We project a substantial net increase in the death burden from neurodegenerative diseases after the mid-century, especially under the unrestricted-emission scenario. These results provide valuable and reliable scientific insights for the field of climate and neurological health, emphasizing the significance of implementing climate policies focused on mitigation and adaptation in an era of aging.

## Results

### Descriptive data and projected temperature

The basic characteristics of neurodegenerative disease deaths are summarized in Supplementary Table 1. In total, 437,218 deaths from overall neurodegenerative diseases were evaluated in the study, including 375,776 deaths from dementias, 65,254 deaths from Alzheimer's disease, 310,522 deaths from non-Alzheimer dementias, and 51,428 deaths from Parkinson's disease. Among these, 92.2% were individuals aged 65 years or older, 53.5% were female, and 91.6% had junior high school education or less. The subtropical monsoon zone and temperate monsoon zone accounted for 62.6% ($N = 273,820$) and 33.4% ($N = 146,033$) of overall neurodegenerative disease deaths, respectively. In contrast, the other three climate zones (temperate continental zone, tropical monsoon zone, highland alpine zone) accounted for 2.9% ($N = 12,724$), 0.5% ($N = 2101$), and 0.6% ($N = 2540$) of overall neurodegenerative disease deaths, respectively. The annual mean temperature was 13.5 °C nationally, with the lowest in the highland alpine zone (mean: 3.7 °C) and the highest in the tropical monsoon zone (mean: 24.4 °C) (Supplementary Table 2). Supplementary Fig. 1 depicts consistently increasing trends in predicted annual-mean temperature relative to the historical period under various climate change scenarios. Nationally, future climate warming is more prominent in the SSP585 scenario (rise of 5.4 °C by 2090s) than the SSP245 scenario (2.9 °C) and the SSP126 scenario (1.9 °C) (Supplementary Table 3). The differences in projected temperatures among the three scenarios become more apparent after the mid-twenty-first century.

### Temperature-death association during observation period

Figure 1 shows exposure–response curves illustrating the relationships between daily mean temperature and death from overall and specific neurodegenerative diseases at the national and climatic zones levels. The curves exhibited similar shapes across various neurodegenerative diseases and climatic zones (subtropical monsoon zone and temperate monsoon zone). Generally, they followed an inverted J-shaped pattern, indicating that both low and high temperatures increased the death risks. The minimum-mortality temperatures ($T_{mm}$) for overall neurodegenerative diseases, dementia, Alzheimer's disease, and non-Alzheimer's dementia were around 24–26 °C, whereas Parkinson's disease had a weakened cold effect and lower $T_{mm}$ (18.3 °C nationally). These $T_{mm}$ values were used as reference temperatures for further analyses (Table 1). There were no clear exposure-response relationships in the temperate continental, tropical monsoon, and highland alpine climate zones (Supplementary Fig. 2), which were thus dismissed in further analyses by climatic zones.

Figure 2 depicts the lag patterns for the relative risks of death from various neurodegenerative disease associated with extreme high temperatures and extreme low temperatures. Generally, for extreme high temperatures, the risks were highest on the current day, persisted for 5 days, and then became insignificant. Conversely, for extreme low temperatures, negative risks were observed on the first two lag days, followed by a significant rise in risks that peaked at lag 6 day, and gradually decreased until lag 14 day. The lag patterns for the effects of high and low temperatures were largely consistent across different diseases and regions.

Table 1 presents the relative risks (RRs) of death from overall and specific neurodegenerative diseases associated with extreme low and high temperatures over lag 0–14 day. The effect of temperature may vary slightly depending on the region and the specific disease. Nationally, the heat-related RRs ranged from 1.29 to 1.41, with higher risks observed for Parkinson's disease. The cold-related RRs ranged from 1.22 to 3.44, with significantly higher risks for Alzheimer's disease and relatively lower risks for Parkinson's disease. In general, the subtropical monsoon zone exhibited higher cold-related death risks than the temperate monsoon zone, while the temperate monsoon zone had higher heat-related death risks.

Supplementary Fig. 3 demonstrates the stratified analysis of overall neurodegenerative disease by age, gender, education level, and combinations of age and sex. Females and individuals with lower education levels faced higher risks of excess death due to extreme temperatures. Regarding the age-stratified results, individuals aged ≥75 were more sensitive to extreme low temperatures, while those aged ≤64 were more susceptible to extreme high temperatures. Older females, in particular, exhibited increased vulnerability to death from neurodegenerative diseases associated with non-optimal temperatures compared to older males.

### Temperature-related death burden due to future climate warming

Figure 3 depicts the national temporal trends of projected attributable fractions (AFs) of neurodegenerative diseases due to heat and cold from the 2010s to 2090s under different climate scenarios. This graph reveals that heat-related neurodegenerative disease deaths would increase, while cold-related deaths would decrease. The slopes are steeper under the high-emission scenario (SSP585), whereas the trends are less steep under the medium-emission scenario (SSP245) and less pronounced under the scenario with strict mitigation strategies (SSP126). The future changing patterns for specific neurodegenerative diseases varied considerably. Cold-attributable fractions of dementia, Alzheimer's disease, and non-Alzheimer's dementia were significantly greater than heat-attributable fractions in the first decades of twenty-first century and would gradually decline with time. In contrast, cold-attributable fraction of Parkinson's disease was minimal, and heat-attributable fraction was much larger and would progressively increase over time. Supplementary Figs. 4 and 5 show largely similar temporal trends of projected AFs of neurodegenerative disease deaths

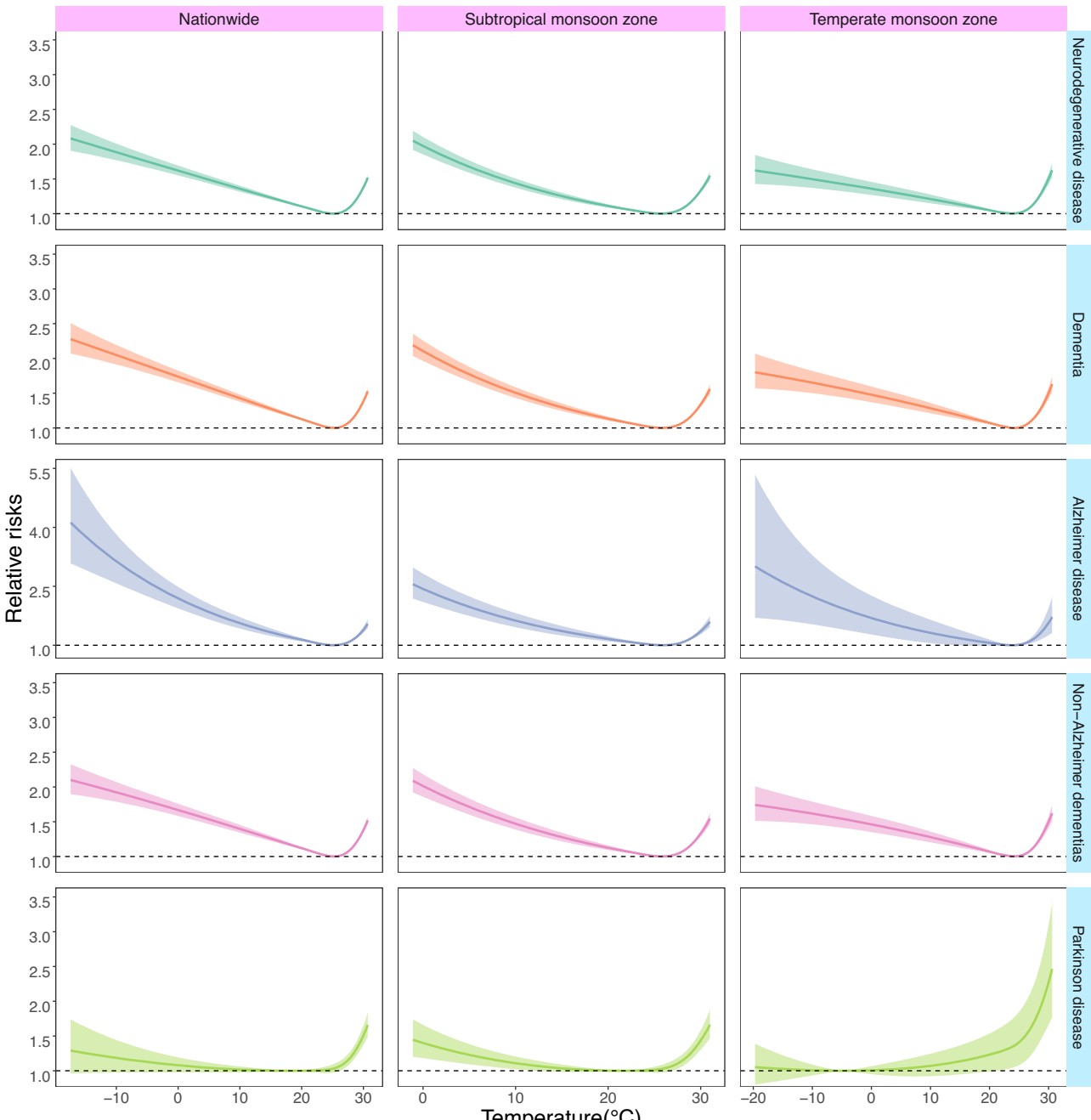

**Fig. 1 | Exposure-response curves for the associations between daily mean temperature and death of overall and specific neurodegenerative diseases, at the national and regional levels.** The associations were presented as the cumulative relative risks comparing a given temperature to the minimum-death temperatures over lag 0–14 day. The lines represent the point estimates, and the shadings indicate corresponding 95% confidence intervals. The results for other climatic zones were not presented due to the null associations. Source data are provided as a Source Data file.

in the subtropical monsoon and temperate monsoon climate zones. The average number of excess deaths from overall neurodegenerative diseases per decade between 1980 and 2009 attributable to cold and heat were 99,560 (95% empirical confidence intervals (eCIs): 83,032–115,909), and 9999 (95% eCI: 7511–13,226), respectively (Supplementary Table 4).

Figure 4 depicts the differences in the predicted death fractions from various neurodegenerative diseases associated with non-optimum temperatures compared with the historical period at the national level. Specifically, in the high-emission scenario, there is a prominent increase in heat-related AF and a significant decrease in cold-related AF throughout the century. In contrast, both heat- or cold-

related AFs show modest variations in the medium- and low-emission scenarios, especially after the middle of this century. Nationally, compared to the historical period, heat-related AFs of overall neurodegenerative diseases, dementia, Alzheimer's disease, non-Alzheimer's dementia, and Parkinson's disease would increase by 2.4%–10.0%, 2.4%–10.2%, 2.7%–10.9%, 2.3%–10.0%, 3.4%–11.4% in the 2090s under the three scenarios, respectively (Supplementary Table 5). The cold-related AF would decrease by 2.4%–5.6%, 2.8%–6.3%, 3.4%–7.5%, 2.6%–6.1%, 0.8%–1.6% in the 2090s under the three scenarios, respectively (Supplementary Table 6). In the 2090s, the number of heat-related deaths of overall neurodegenerative diseases will increase by 15,384 (95% eCI: 6455–26,185) under SSP126, 26,371 (95% eCI:

**Table 1 | The cumulative relative risks (means and 95% confidence intervals) of neurodegenerative diseases death comparing extreme high temperatures (the 97.5ᵗʰ percentile) and extreme low temperatures (the 2.5ᵗʰ percentile) to the minimum-death temperatures over lag 0–14 day**

| Region | Neurodegenerative disease | Dementia | Alzheimer disease | Non-Alzheimer dementias | Parkinson disease |
|---|---|---|---|---|---|
| **Nationwide** | | | | | |
| MMT (°C) | 25.1 | 25.3 | 25.2 | 25.2 | 18.3 |
| P2.5 (−12.5 °C) | 1.95 (1.81, 2.10) | 2.13 (1.96, 2.30) | 3.44 (2.73, 4.33) | 1.98 (1.82, 2.16) | 1.22 (0.96, 1.54) |
| P97.5 (29.5°C) | 1.29 (1.26, 1.32) | 1.29 (1.26, 1.33) | 1.30 (1.23, 1.38) | 1.29 (1.25, 1.33) | 1.41 (1.28, 1.56) |
| **Subtropical monsoon zone** | | | | | |
| MMT (°C) | 25.8 | 25.9 | 26 | 25.9 | 21.4 |
| P2.5 (1.5 °C) | 1.88 (1.76, 2.00) | 2.00 (1.86, 2.14) | 2.28 (1.97, 2.64) | 1.92 (1.77, 2.08) | 1.34 (1.15, 1.57) |
| P97.5 (29.9 °C) | 1.31 (1.27, 1.35) | 1.32 (1.27, 1.36) | 1.33 (1.25, 1.42) | 1.31 (1.26, 1.36) | 1.41 (1.27, 1.57) |
| **Temperate monsoon zone** | | | | | |
| MMT (°C) | 23.9 | 24.2 | 23.9 | 24.2 | −4.8 |
| P2.5 (−15.3 °C) | 1.57 (1.40, 1.75) | 1.74 (1.54, 1.96) | 2.63 (1.62, 4.25) | 1.69 (1.49, 1.91) | 1.03 (0.85, 1.24) |
| P97.5 (29.2 °C) | 1.36 (1.29, 1.43) | 1.35 (1.28, 1.43) | 1.41 (1.17, 1.69) | 1.35 (1.28, 1.43) | 2.05 (1.50, 2.78) |

*MMT* minimum-death temperatures, *P2.5* 2.5ᵗʰ percentile, *P97.5* 97.5ᵗʰ percentile.

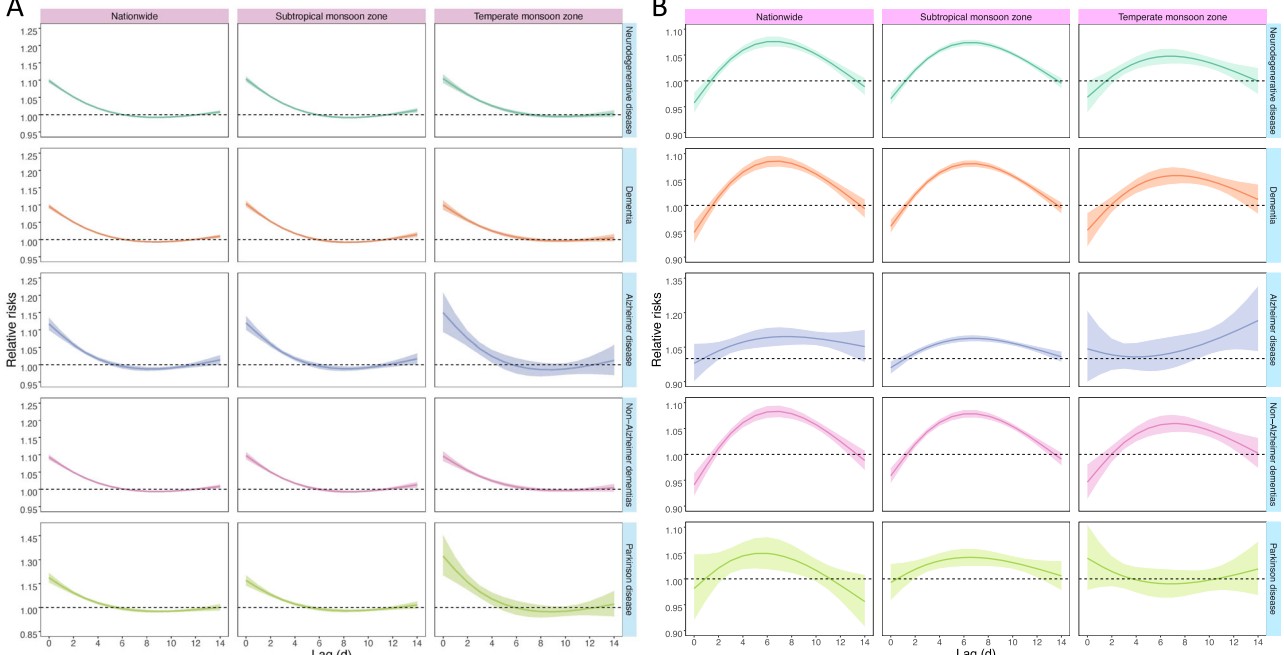

**Fig. 2 | Lag-response curves for the relative risks of neurodegenerative disease death comparing extreme high temperatures (the 97.5ᵗʰ percentile, A) and extreme low temperatures (the 2.5ᵗʰ percentile, B) to the minimum-death temperatures, at the national and regional levels.** The lines represent the point estimates, and the shadings indicate corresponding 95% confidence intervals. The results for other climatic zones were not presented due to the null associations. Source data are provided as a Source Data file.

13,744–39,683) under SSP245, and 69,946 (95% eCI: 31,899–114,617) under SSP585, while the number of cold-related deaths of overall neurodegenerative diseases will decrease by 15,734 (95% eCI: 21,428–9749) under SSP126, 22,569 (95% eCI: 33,774–15,517) under SSP245, and 41,285 (95% eCI: 62,950–28,147) under SSP585 (Supplementary Tables 7 and 8).

With regard to the net changes in death fractions associated with future heat and cold, there appears to a net small decrease or no net change in the first decades of this century, followed by a net increase after the mid-twenty-first century, particularly in the high-emission scenario (Fig. 4). Nationally, the net increase in AFs of death from all neurodegenerative diseases would be 0.7% in the 2050s, and 4.5% in the 2090s under the high-emission scenario (Supplementary Table 9). In terms of the net changes in attributable number, there would be an

increase of 3569 in the 2050s and 28,715 in the 2090s nationally under the high-emission scenario (Supplementary Table 10). Supplementary Figs. 6 and 7 illustrate similar patterns of net changes in death fractions associated with future heat and cold in the subtropical monsoon and temperate monsoon zones. The magnitude of net changes differed significantly across specific diseases and climatic zones (Supplementary Tables 9 and 10). The largest net increases in AFs were found for Parkinson's disease and in the temperate monsoon zone.

## Discussion
This nationwide, individual-level, case-crossover study demonstrated significant relationships between daily mean temperature and death from multiple neurodegenerative disorders with increased death risks for both low and high temperatures. The excess risks were

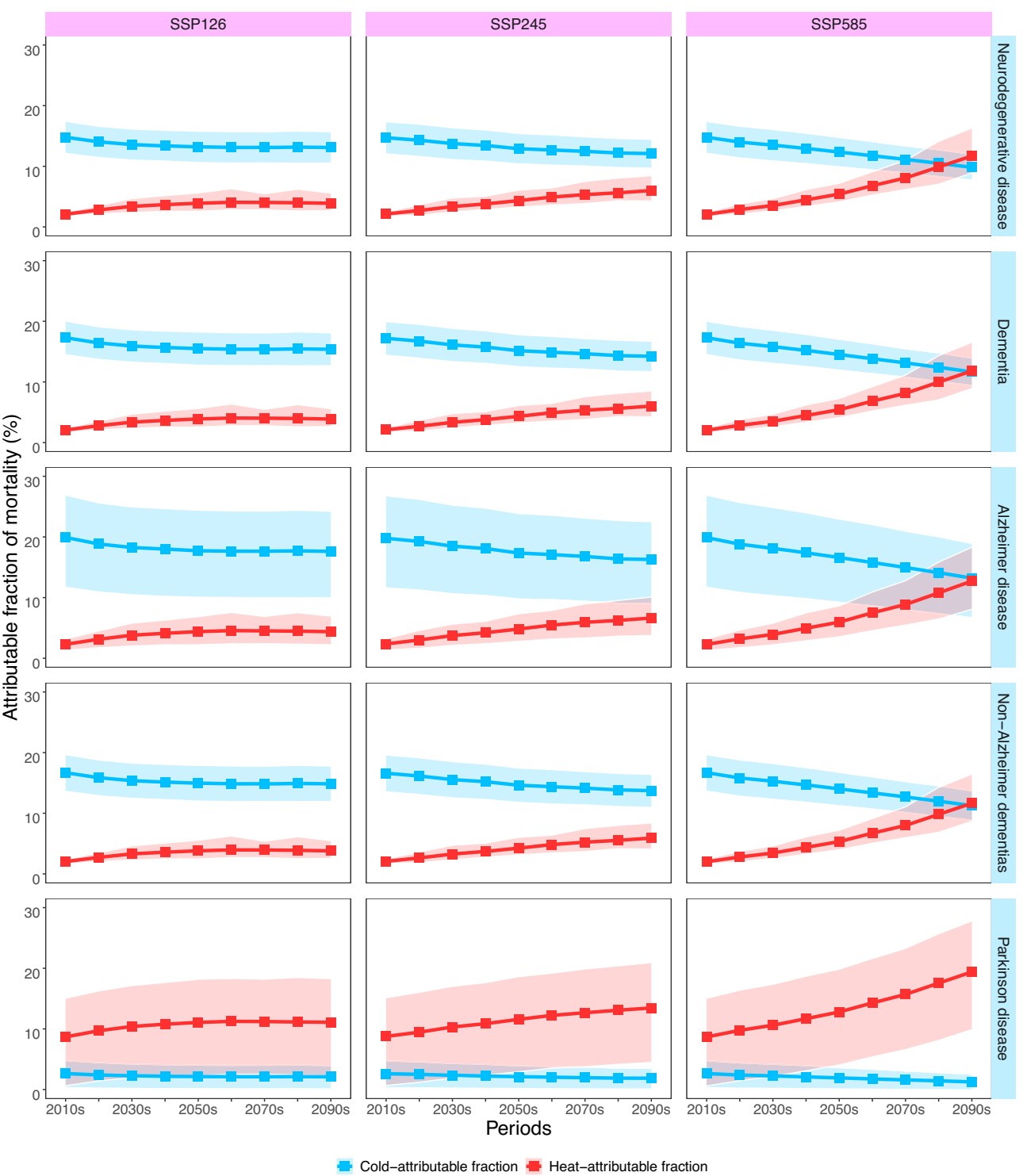

**Fig. 3 | The projected national fractions of neurodegenerative disease death associated with non-optimum temperatures under three climate change scenarios (SSP126, SSP245 and SSP585) for every decade from the 2010s to 2090s.** Estimates are reported as GCM-ensemble averages. The points denote the mean estimates and the shaded areas represent their empirical 95% confidence intervals computed from Monte Carlo simulations (1000 samples). SSP shared socio-economic pathway, GCM general climate models. Source data are provided as a Source Data file.

more prominent among the elderly, females, and those with low education levels. We projected that heat-related deaths from neurodegenerative diseases will increase, while cold-related deaths will decrease throughout the century. These trends are more pronounced in the high-emission scenario but less pronounced in scenarios with mitigation strategies. There would be a net increase in death associated with non-optimal temperatures after the

mid-twenty-first century, especially under the unrestricted-emission scenario. The magnitude of net changes differed significantly across specific neurodegenerative diseases and climatic zones. These findings provide valuable and reliable scientific knowledge for the field of climate and neurological health, emphasizing the importance of implementing mitigation and adaptation climate policies in an aging society.

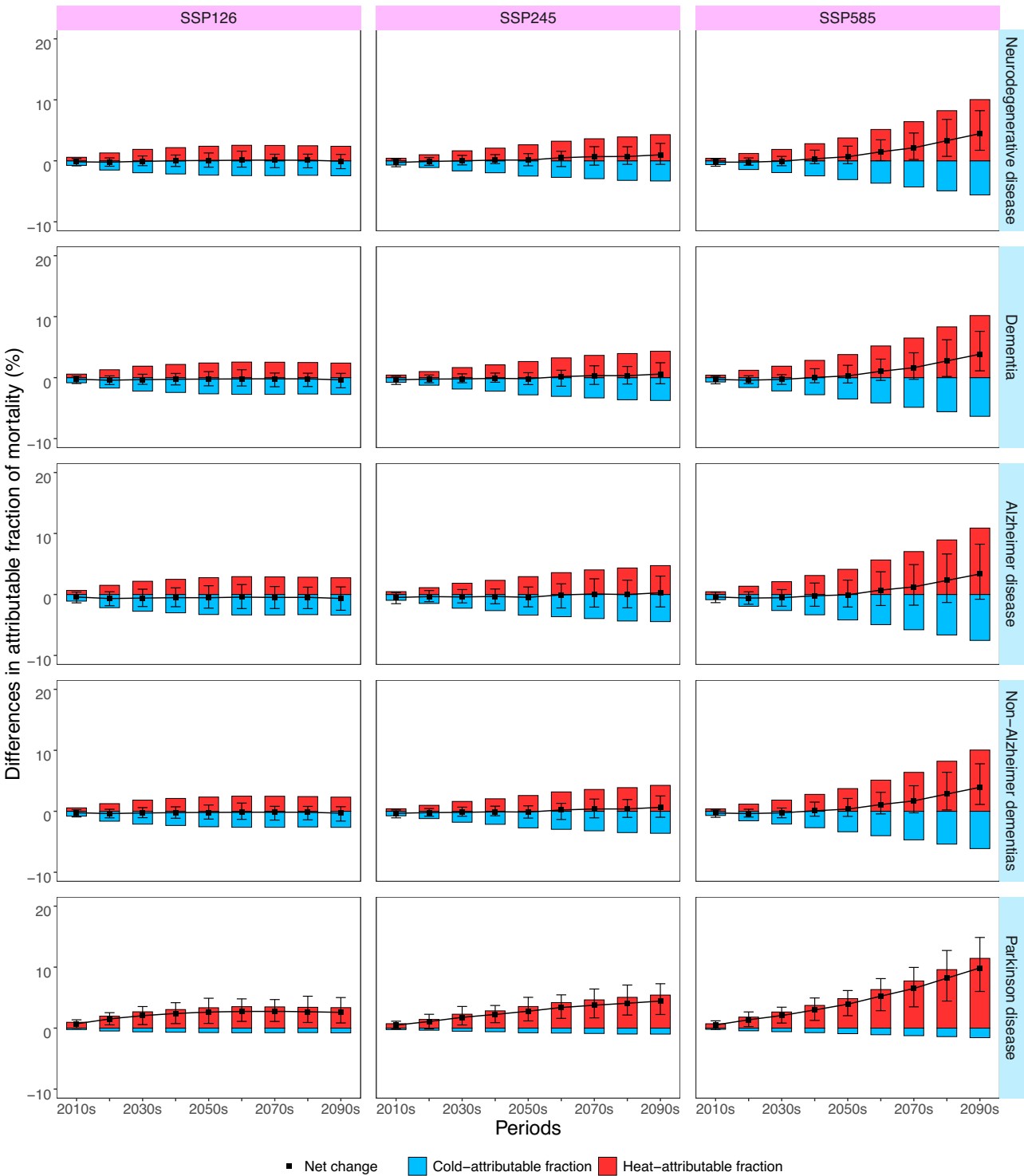

**Fig. 4 | Differences in the projected national fractions of neurodegenerative disease death associated with non-optimum temperatures in 2010–2090 compared with 1980–2009 under three climate change scenarios (SSP126, SSP245 and SSP585).** Estimates are reported as GCM-ensemble averages. The red bars represent the estimates of the heat-attributable fraction and the blue bars represent the estimates of the cold-attributable fraction. The black dots and vertical segments represent estimates of net differences and their 95% empirical CIs, respectively. SSP shared socioeconomic pathway, GCM general climate models. Source data are provided as a Source Data file.

We found a consistent and inverse J-shaped association between daily mean temperature and death from various neurodegenerative diseases. The effects of cold were found to persist for a longer duration compared to the effects of heat. These findings align with existing epidemiological literature linking ambient temperature to a range of adverse health outcomes, including neurodegenerative diseases[15,21,22].

However, previous studies have primarily focused on associating heat with overall or specific neurodegenerative diseases, while the effects of cold have been understudied and yielded inconsistent conclusions[16–18,23]. Utilizing the largest and most comprehensive death dataset, our study demonstrates that both cold and heat are associated with increased risks of death from neurodegenerative diseases.

The present study further evaluated four main neurodegenerative diseases (dementia, Alzheimer's disease, non-Alzheimer's dementia, and Parkinson's disease) in different climatic zones. Our analysis revealed that the $T_{mm}$ for the temperature-death association were largely comparable for various specific neurodegenerative diseases (24–26 °C) except for Parkinson's disease (18.3 °C). The heat-related RRs were appreciably larger for Parkinson's disease, whereas the cold-related RRs were higher for Alzheimer's disease. Although the exact mechanisms for these differences were unclear, our results provide valuable epidemiological evidence on the differential response of various neurodegenerative diseases to non-optimum temperatures. This information is crucial for informing public health interventions. The subtropical monsoon zone exhibited a higher risk of cold-related deaths compared to the temperate monsoon zone, potentially due to weaker cold adaptation in the former and lower heat adaptation in the latter[22,24]. We also identified potential effect modification, which is important for the development of evidence-based health protection plans to address climate change-related adverse health risks. We found that extreme temperatures were associated with greater risks for the elderly, females and those with lower education levels, potentially indicating the vulnerability of these subgroups[25].

Although the physiological mechanisms underlying how climate change affects death from neurodegenerative disorders are not well understood, several factors may be involved, including oxidative stress, excitotoxicity, neuroinflammation, apoptosis, and changes in blood flow and brain metabolism[16–18,23,26]. Non-optimal ambient temperatures have been shown to contribute to acute cognitive impairment and long-term deterioration in patients with neurodegenerative diseases[12–14]. Climate change may induce neurodegenerative diseases through increased oxidative stress, leading to neuroinflammation and subsequent neuronal and glial apoptosis. This process can also promote the formation of neurofibrillary tangles and amyloid plaque deposition[7,27–29]. These molecular mechanisms could act in individuals with neurogenic diseases or in older adults with impaired thermoregulation, thereby increasing the exacerbation and death risks of neurodegenerative diseases. Further, temperature alterations may influence cerebral blood flow and metabolism, which are crucial for maintaining brain health[30,31]. High temperatures may result in vasodilation and increase blood flow, leading to cerebral hyperperfusion[32]. In contrast, low temperatures may induce vasoconstriction and reduce blood flow, potentially impairing cerebral metabolism and exacerbating neurodegenerative processes[31,32]. However, the effects of temperature could vary across different diseases due to their distinct pathophysiological characteristics. Notably, both high and low temperatures are associated with increased risks of Alzheimer's disease, albeit with potentially different underlying mechanisms. High temperatures are thought to promote amyloid-β aggregation and induce neuronal damage[25,33]. Conversely, low temperatures can impair cognitive function and worsen pathological changes linked to Alzheimer's disease by reducing cerebral blood flow and metabolism[34]. Regarding Parkinson's disease, the impact of high temperatures is more pronounced, while there is a weaker association with low temperatures, likely due to the exacerbation of oxidative stress and inflammation[28,35]. Nevertheless, due to the complexities and intricacies involved in understanding the specific temperature effects on different neurodegenerative diseases, further research is necessary to delve deeper into these relationships.

Future changes in death burden from various neurodegenerative disorders associated with non-optimal temperatures have rarely been investigated in the context of climate warming. A previous time-series study in England estimated that by 2040, heat-related hospital admissions of dementia (F00-F03) would increase by nearly 300% compared to the baseline under a high emission scenario[20]. However, this study only examined the heat-related burden of dementia admissions until 2040, considering two climate scenarios. In our study, we utilized nonlinear and lagged temperature-death relationships to project the death burden associated with non-optimal temperatures for various neurodegenerative diseases in different climate zones under multiple climate change scenarios. We predicted that heat-related AFs for all specific neurodegenerative diseases would increase with warming, while cold-related AFs would decrease, characterized by a steeper slope in the high-emission scenario. There would be a net increase of death burden due to non-optimal temperatures after the mid of this century, especially under the unrestricted-emission scenario. This is plausible considering that unrestricted carbon emissions would lead to the greatest magnitude of climate warming, resulting in a disproportionately high proportion of heat-related excess deaths. We further projected that the death AFs for Parkinson's disease would experience the largest net increase, especially in the temperate monsoon climate zone. This may be attributed to the heightened impact of hot temperatures on Parkinson's disease death than cold temperatures, as well as the worse adaptation ability to hot weather for people in the temperate monsoon climate zone. We contributed to compelling evidence for the differential changing trends for the death burden of various neurodegenerative diseases, which is valuable for design effective public health strategies to address these challenges in the context of climate change. Our study provides clear evidence that future warming will exacerbate the overall burden of death from various neurodegenerative disorders, unless strict climate policies were adopted.

The limitations of our study should be acknowledged. First, although this study covers all areas of China Mainland, our results for the temperate continental zone, the tropical monsoon zone, and the high-altitude alpine zone were of high uncertainty due to the very limited sample size. Accordingly, our findings could not be generalized to these climatic zones or to other countries with different socioeconomic characteristics. Second, our projections relied on the current estimates of temperature-death relationships and did not take into account the effects associated with the changes in demographic features and population adaptation that may complicate our estimations. Third, as temperature data are derived from ambient measurements rather than individual monitoring, exposure measurement errors are inevitable and may have underestimated our results. Fourth, we did not conduct analyses at the local (e.g., county) level due to the small numbers of daily neurodegenerative deaths. This may impact the risk estimates for extreme temperatures due to the variations in local socio-economic, demographic, and environmental characteristics[36,37], but could not impact the shape of the exposure-response curves nor the assessment of disease burden due to overall non-optimum temperatures. Fifth, the scope of our study was limited to the deceased population. While death data offers valuable insights into the overall disease burden, it does not encompass the entire population affected by the disease or fully capture the extent of the disease burden. Sixth, despite rigorous data verification and audit procedures, diagnostic errors may occur due to the complexity and heterogeneity of neurodegenerative diseases.

In summary, this nationwide, individual-level case-crossover study offers convincing evidence that both high and low temperatures are associated with increased risks of death from multiple neurodegenerative diseases. We further demonstrated that climate warming will lead to a significant increase in heat-related death burden of neurodegenerative diseases and a considerable decrease in cold-related death burden. There would be a notable net increase of death burden from neurodegenerative diseases after the mid of this century, especially for Parkinson's disease and in the unrestricted-emission scenario. These results underscore the need to implement effective climate and public health policies to mitigate the mounting challenges for neurodegenerative diseases associated with global warming.

## Methods

The Institutional Review Board at the School of Public Health, Fudan University deemed the study exempt from review and waived the requirement for informed consent because the study involved analysis of deidentified data. Our study adhered to the Strengthening the Reporting of Observational Studies in Epidemiology (STROBE) reporting guideline.

### Study design

We employed the individual-level, time-stratified, case-crossover approach to investigate the associations between ambient temperature and death from overall and specific neurodegenerative diseases at the national level and across different climatic zones. Compared to the traditional ecological time-series that based on daily aggregate health data, this kind of design could minimize the confounding effect of all individual-level, time-invariant risk factors through a self-matching strategy and automatically excludes temporal trends (e.g., seasonality) by selecting controls within a month[38,39]. Afterwards, we projected future excess deaths of neurodegenerative diseases associated with non-optimal temperatures at the county level. Using the temperature-death associations, daily temperature simulations from ten general circulation models (GCMs) for the historical period (1980–2009) and the scenario period (2010–2099) were converted into the projections of excess deaths associated with non-optimum temperatures. To accurately assess the net changes in the burden of neurodegenerative disease death due to climate change, we separately predicted excess deaths associated with cold and heat throughout the historical and scenario periods.

### Data collection

We used death data from the China Cause of Death Reporting System (CDRS), a well-established system designed by the central government to collect information from all deaths occurring in China Mainland. The data collection process adhered to stringent protocols, standard procedures, and meticulous quality control measures. Regular training and supervision were carried out at all administrative levels, from township to national, throughout the year to assure the quality of each death certificate reported. Experienced staff in each district- or county-level Center of Disease Control and Prevention assigned and coded the underlying cause of death using the International Classification of Diseases (ICD) coding system. The data from this system are widely used by the central government to produce official mortality statistics for informing health policy, as well as being a reliable data source for scientific research. Detailed descriptions of the registry were published elsewhere[40,41]. We collected individual death records of neurodegenerative diseases in all 2844 county-level administrative areas of China Mainland from 2013 to 2019. As per the 10th edition of International Classification of Diseases, we derived the data of overall neurodegenerative diseases, dementia, Alzheimer's disease, non-Alzheimer's dementia, and Parkinson's disease according to the underlying cause of death (Supplementary Table 1). We also gathered information on demographics (e.g., gender, age, education level), date of death, and residential address.

We derived daily temperature and relative humidity data from the fifth generation atmospheric reanalysis product (ERA5), which was developed by the European Centre for Medium-Range Weather Forecasting and has high spatial and temporal resolution[42]. The ERA5-Land component of the dataset produces temperature series that closely align with station measurements, employing data assimilation techniques[43]. We extracted hourly temperature series for all counties from the nearest grid cell in the ERA5-Land dataset. Subsequently, we calculated the daily mean temperature by averaging all 24-h temperature estimates for each grid cell.

### Projected temperature data under climate change scenarios

The projected temperature data were obtained from the NASA Earth Exchange Global Daily Downscaled Projections dataset. The downscaled products are produced using daily variants of the monthly bias correction/spatial decomposition method, with a horizontal resolution of ¼ degree[44]. We extracted daily mean temperature from ten GCMs for the historical period and the projection period under three new climate change scenarios (SSP126, SSP245, and SSP585) (Supplementary Table 11). These scenarios illustrate corresponding strategies for climate adaptation and mitigation under specific shared socioeconomic pathways (SSPs) to accomplish the objectives of representative concentration pathways (RCPs)[45]. Specifically, SSP126 and SSP245 depict a moderate scenario with strict carbon reduction and a medium scenario with partial carbon reduction, respectively, while SSP585 represents an extreme scenario with no mitigation policies. We extracted simulated daily temperature series for each county for the period 1980–2099. The obtained temperature series were then calibrated with the ERA5 data using the bias-correction method[46]. This method preserves the original data's trends and variability by adapting the cumulative distribution of the simulated data to the observed data[46].

### Statistical analysis

**Estimation of the temperature–death association.** For the time-stratified case-crossover approach, we utilized conditional logistic regression models combined with distributed lagged nonlinear models (DLNM) to estimate the temperature-death associations. This analysis was conducted at both the national level and within different climatic zones, including the subtropical monsoon zone, temperate monsoon zone, temperate continental zone, tropical monsoon zone, and highland alpine zone.

To establish the exposure-response relationship, we incorporated a cross-basis function of daily mean temperature provided by the DLNM into the conditional logistic regression model. Model selection was based on the Akaike Information Criterion to ensure optimal parameter fit. Meanwhile, to ensure consistency and comparability across regions and outcomes, we employed a uniform lag dimension for the primary analysis. We used natural cubic splines with three internal knots located at the 10th, 75th, and 90th percentiles of the national or climatic zone-specific daily temperature distributions during the observation period (2013–2019) to capture potential nonlinear exposure-response relationships. The lag-response relationships were modeled using three internal knots placed at equal intervals on a logarithmic scale, with a maximum lag of 14 days. To account for potential confounding effects, the model additionally included a categorical variable for holiday and a natural cubic B spline with the degrees of freedom (*df*) of 3 for relative humidity. Throughout the analysis, we utilized all available data to establish exposure-response relationships for the complete temperature series. When plotting the exposure-response relationship, we restricted the display to the 1st to 99th percentiles of the temperature series to mitigate statistical uncertainty and wide confidence intervals at extreme exposures due to small sample size.

We reported the RRs and their 95% CIs for death from overall and specific neurodegenerative diseases at extreme high temperatures (the 97.5th percentile) and extreme low temperatures (the 2.5th percentile) compared to the reference temperatures[47,48], which were calculated as $T_{mm}$ (which is the temperature that corresponds to the lowest mortality risk) in the respective exposure-response curves.

To explore potential effect modifications, we performed stratified analyses for overall neurodegenerative diseases at the national level, stratifying by age (≤64, 65–74 and ≥75), gender (male vs. female), education level (middle school or below vs. high school or above), and combinations of age and sex (age <65 and Male, age ≥65 and <74 and

Male, age ≥75 and Male, age <65 and Female, age ≥65 and <74 and Female, age ≥75 and Female).

**Projections of death burden associated with future non-optimum temperature.** Based on the corresponding nonlinear and lagged temperature-death relationships for specific climatic zones, we calculated the number of excess deaths associated with non-optimum temperatures ($D_{attr}$) for each county and for each combination of GCMs and SSPs during 1980 to 2099 according to the following formula[5,49]:

$$D_{attr} = D \cdot \left( 1 - e^{-(f^*(T^*_{proj};\theta^*_b) - f^*(T^*_{mm};\theta^*_b))} \right)$$

where $D$ denotes the average daily number of deaths for the observation period (2013–2019). $f$ and $\theta^*$ represent the uni-dimensional overall cumulative exposure–response curves with reduced lag dimension, derived from the bi-dimensional term estimated in the section headed "Estimation of the temperature–death association" of the method. $T^*_{pro}$ corresponds to the projected temperature series. We computed the daily attributable deaths as GCM-ensemble averages by aggregating by county, decade, and SSP. We then calculated the AF of death due to non-optimum temperatures as the proportion of excess deaths for each county, decade, and SSP in the corresponding total deaths, which were computed using the baseline death by averaging the observed daily series from 2013 to 2019. We separately calculated excess deaths due to heat and cold by summing the subsets corresponding to days with temperatures higher or lower than $T_{mm}$ to compute heat- or cold- related AF. Finally, we calculated the differences of AFs in each decade of the scenario period (the 2010s, 2020s,..., and 2090s) compared with those in the historical period (i.e., the average of AFs in each decade between 1980 and 2009), which represents the death burden due to future climate warming. We also calculated the differences of projected excess deaths in each decade of the scenario period (the 2010s, 2020s,..., and 2090s) compared with those in the historical period. To quantify uncertainty, we used Monte Carlo simulations to obtain 95% eCIs for both the estimation of exposure-lag-response relationships and climate projections across GCMs. Detailed methodological steps can be found in the published hands-on tutorial[49].

All statistical analyses were performed using R software (version 3.6.1, R Project for Statistical Computing) with the "dlnm" and "survival" package. All analyses were two sided with an alpha of 0.05.

**Reporting summary**
Further information on research design is available in the Nature Portfolio Reporting Summary linked to this article.

## Data availability

All data supporting the findings described in this manuscript are available in the article and in the Supplementary Information. The data generated in this study are available under restricted access for the identifiable nature of the data and data management requirements. Access can be obtained by contacting the corresponding author (kanh@fudan.edu.cn) and will be answered within 12 weeks. This study utilized death data from the China Cause of Death Reporting System (CDRS), which is not publicly available due to a restricted data use agreement with the national institute. Meteorological data were sourced from the fifth generation atmospheric reanalysis product (ERA5), accessible at https://cds.climate.copernicus.eu/cdsapp#!/search?type=dataset. The projected temperature data were obtained from the NASA Earth Exchange Global Daily Downscaled Projections dataset, available from https://www.nasa.gov/nex/gddp. Source data are provided with this paper.

## Code availability

The custom code supporting the findings of this study is available upon request from the corresponding author (kanh@fudan.edu.cn).

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

## Acknowledgements

We thank all staff in the 31 provincial Centers for Disease Control and Prevention (CDC), and all other local CDCs in Disease Surveillance Point Systems for assisting with data collection, cleaning, and management. This work is supported by the National Natural Science Foundation of China (92043301, H.K.), National key research and development program (2022YFC3702701, R.C.) and the Shanghai Municipal Science and Technology Commission (21TQ015, R.C.).

## Author contributions

P.Y., Y.G. and R.C. are joint first authors. H.K., M.Z. and J.H. contributed equally to the correspondence work. H.K., M.Z. and J.H. contributed to the conceptualization of the study, methodology, funding acquisition, validation, review, editing and supervision of the manuscript. P.Y., Y.G. and R.C. contributed to the data curation, formal analysis, methodology, visualization, writing the original draft, and review and editing of the manuscript. C.H. and W.L. also contributed to the data curation. All authors critically reviewed the manuscript and approved the final manuscript.

## Competing interests

The authors declare no competing interests.
