## [Peer Review File · Nature Communications]

Reviewers' Comments:

Reviewer #1:

Remarks to the Author:

It is a study that analyses "Temperature-related mortality burden of various neurodegenerative diseases under climate warming: a nationwide modelling study". Although it is an interesting and novel study, methodologically it has a serious problem. This consists of using standard percentile values: "extreme high temperatures (the 97.5 th percentile) and extreme low temperatures (the 2.5th percentile)".

Definitions of extreme temperatures should be determined for each location through epidemiological studies as established by the WHO. Socio-economic, demographic and local factors may cause the heat definition temperature to correspond to a 99th percentile in one place and a 90th percentile in another, so that what would be a heat wave in one place would not be a heat wave in another and different situations would be compared. An extreme temperature in health cannot be defined from an exclusively climatic point of view without taking into account local factors.

Reviewer #2:

Remarks to the Author:

Review of NCOMMS-23-10912

The manuscript presents an assessment of temperature-related neurodegenerative deaths and projections under climate change scenarios. The association with temperature seems clear-cut and the results also show an expected net increase in future climate.

This is a nice and well conducted study that uses all the recent statistical machinery of environmental epidemiology in an appropriate manner. The large scope also makes it an important study I would say.

I have several comments, although none of them represent any major flaw in the paper.

Major (sort of)

1. A very general consideration first. The authors claim that this kind of study is important given that neurodegenerative diseases are important DALYs. However, the outcome of the study is not really the apparition or incidence of these diseases, but deaths of people suffering from these diseases.

My question is then, can these results simply represent temperature-related mortality, regardless of whether deceased were suffering from one of these diseases, it just happens that the sample is restricted to this population?

2. I am struggling making sense of the lag-response functions shown in Figure 2B. Most of them present RRs below 1 at lags 0-1 and some of them important RRs at the end.

This makes me think that the lag dimension of the DLNM is perhaps not optimally set. It seems to me that the specification is roughly the one usually used in all-cause mortality, but perhaps in this context this is not the best one. Have any kind of model selection been performed by e.g. comparing the AIC or by performing sensitivity analyses to the lag-response curve?

3. I would be more cautious in claiming a causal association simply because a case-crossover design has been used here. My understanding is probably incomplete but I think case-crossover and time-series studies are roughly equivalent in the environmental epidemiology context (see <https://doi.org/10.1093/biostatistics/kxl013>). I think one way to see this is that the individual confounders that are controlled for by design are not really confounders in time series designs anyway, since we don't expect them to vary across time, or they vary they will mostly be captured by time-related terms in the models. Additional information on causality in the case-crossover design can be found in this nice paper: <https://doi.org/10.1111/biom.13749>.

Finally, note that the paper cited by the authors when claiming causality ([33] l. 307) does not really talk about that but is just another application of this design, and thus irrelevant here.

4. We don't get a good sense of the public health burden associated with temperature-related neurodegenerative diseases as no measure of prevalence is given. I think it would be interesting to either report attributable numbers, or death rates somewhere. No need to revolutionise the paper for this, but a few figures in the text would be relevant.

Minor

I. 97: 0.43 million, so 430,000? Please use the same formatting as the following figures.

I. 100: does it represent a disproportionate amount given the population of each climate zone? Also, if other zones account for only 4% (each?), where are the remaining ~25%?

Not sure both Figure 3 and 4 are needed as they roughly provide the same information.

Higher risks in females can be explained by different age structure. Does the result hold when it is stratified by both age and sex?

I. 366 Truncating the extremes of temperature is a bit odd as these extremes could also have important effects. A sensitivity analysis to this would be interesting.

Methods: I guess Tmm represents the MMT, but it should be defined more explicitly for less familiar audiences.

Reviewer #3:

Remarks to the Author:

This paper on temperature-related mortality from neurodegenerative diseases adds important epidemiological evidence of climate impacts on health. The statistical power with large number of cases and the case-crossover design minimizing potential confounding are key advantages.

The inverse J-curve is not surprising and mirrors the relationship with all-cause mortality. Also, consistent with other temperature mortality studies, warmer climates have higher Min Mortality Temp (MMT) suggesting some physiological or behavioral adaptation.

I think the authors overstate a causal relationship, mostly pointed to their case cross-over design (which is a strong design). I would like the authors to go further in explaining the robustness of the data going into this study, for example:

1) How reliable is death certificate data in China? Can they verify that the main cause of death is neurodegenerative disease for these death certificates?

2) Related to (1) I think to be more credible, for the same locations (national and the 2 regions) we should see the all-cause mortality graphs. How different are they? Clearly these diseases are covariate with age (no surprise that women show higher response).

3) A stronger discussion on potential physiological explanation would be useful. True that oxidative stress can always be a factor for many diseases. The authors in the discussion could offer more information, for example, on the near opposite relationships to temperature between Alzheimers and Parkinson diseases.

Dear Editors and Reviewers,

Ref.: Ms. No. NCOMMS-23-10912

We thank the editors and reviewers for their thorough reviews, constructive comments, and enthusiastic supports. We have revised the manuscript accordingly and provided point-to-point replies to these comments.

Editorial Requirements:

Reply: All the editorial requirements have been addressed accordingly. The authors declare that there is no conflict of interest.

Reviewer Comments

Reviewer #1:

It is a study that analyses "Temperature-related mortality burden of various neurodegenerative diseases under climate warming: a nationwide modelling study". Although it is an interesting and novel study, methodologically it has a serious problem. This consists of using standard percentile values: "extreme high temperatures (the 97.5th percentile) and extreme low temperatures (the 2.5th percentile)".

Definitions of extreme temperatures should be determined for each location through epidemiological studies as established by the WHO. Socio-economic, demographic and local factors may cause the heat definition temperature to correspond to a 99th percentile in one place and a 90th percentile in another, so that what would be a heat wave in one place would not be a heat wave in another and different situations would be compared. An extreme temperature in health cannot be defined from an exclusively climatic point of view without taking into account local factors.

Reply: We appreciate the reviewer's attention to methodological considerations.

We understand that local factors, including socio-economic, demographic, and environmental considerations, may influence the definition of extreme temperatures for specific locations. Studying the socio-economic, climatic, and environmental characteristics of specific locations enables the identification of potential vulnerability patterns among populations.¹⁻². However, incorporating such factors into our analysis

on a nationwide scale would introduce significant complexities and difficulties due to data limitations.

Notably, within the realm of epidemiological studies, particularly those adopting time-series or case-crossover study designs, **it is most common to employ percentile-based definitions** to characterize extreme temperatures and their health impacts at the national or regional level¹⁻⁵. These percentile values allow for **consistent and comparable assessments of temperature extremes across different locations and time periods**. We have made sure to provide appropriate citations to support this approach in the revised manuscript.

Our study examined the relationship between ambient temperature and death from neurodegenerative diseases across the nation, encompassing five distinct climatic zones. This analysis considered the diverse socioeconomic and environmental factors in different locations, **as areas within the same climate zone generally share similar climatic and environmental conditions**. Accordingly, in projecting future neurodegenerative diseases death burdens attributable to climate warming, **we used corresponding exposure-response relationships for specific climatic zones**, and aggregated the county-specific burdens to derive the national projections. We did not conduct the above analyses at the local (e.g., county) level due to the small numbers of daily neurodegenerative deaths for single locations.

Most importantly, the definition of extreme temperatures only impacts the risk estimates for only a given temperature, but could not impact **the shape of the exposure-response curves or the assessment** of disease burden due to overall non-optimum temperatures. This is because that throughout the analysis, we **incorporated all available data for modeling analysis** to establish exposure-response relationships for the complete temperature series, which were then utilized in the prediction study.

In the Limitations section, we acknowledge that variations in socio-economic, demographic, and local characteristics could result in varied thresholds for extreme temperature events in different locations. Please refer to lines 315-320.

“Fourth, we did not conduct analyses at the local (e.g., county) level due to the small numbers of daily neurodegenerative deaths. This may impact the risk estimates for extreme temperatures due to the variations in local socio-economic, demographic, and environmental characteristics, but could not impact the shape of the exposure-response curves nor the assessment of disease burden due to overall non-optimum temperatures.”

References:

1. Sera F, Armstrong B, Tobias A, et al. How urban characteristics affect vulnerability to heat and cold: a multi-country analysis. *Int J Epidemiol.* 2019;48(4):1101-1112.
2. Masselot P, Mistry M, Vanoli J, et al. Excess mortality attributed to heat and cold: a health impact assessment study in 854 cities in Europe. *Lancet Planet Health.* 2023;7(4):e271-e281.
3. Gasparri A, Guo Y, Hashizume M, et al. Mortality risk attributable to high and low ambient temperature: a multicountry observational study. *Lancet.* 2015;386(9991):369-375.
4. Chen R, Yin P, Wang L, et al. Association between ambient temperature and mortality risk and burden: time series study in 272 main Chinese cities. *BMJ.* 2018;363:k4306.
5. Chen J, Gao Y, Jiang Y, et al. Low ambient temperature and temperature drop between neighbouring days and acute aortic dissection: a case-crossover study. *Eur Heart J.* 2022;43(3):228-235.
6. Jiang Y, Hu J, Peng L, et al. Non-optimum temperature increases risk and burden of acute myocardial infarction onset: A nationwide case-crossover study at hourly level in 324 Chinese cities. *EClinicalMedicine.* 2022;50:101501.
7. Zhou Y, Gao Y, Yin P, et al. Assessing the Burden of Suicide Death Associated With Nonoptimum Temperature in a Changing Climate. *JAMA Psychiatry.* 2023;80(5):488-497.

Reviewer #2:

The manuscript presents an assessment of temperature-related neurodegenerative deaths and projections under climate change scenarios. The association with temperature seems clear-cut and the results also show an expected net increase in future climate.

This is a nice and well conducted study that uses all the recent statistical machinery of

environmental epidemiology in an appropriate manner. The large scope also makes it an important study I would say.

I have several comments, although none of them represent any major flaw in the paper.

Reply: We appreciate the reviewer's generous comments and useful suggestions. We have revised our manuscript accordingly. Please see the following detailed response.

Major (sort of)

1. A very general consideration first. The authors claim that this kind of study is important given that neurodegenerative diseases are important DALYs. However, the outcome of the study is not really the apparition or incidence of these diseases, but deaths of people suffering from these diseases.

My question is then, can these results simply represent temperature-related mortality, regardless of whether deceased were suffering from one of these diseases, it just happens that the sample is restricted to this population?

Reply: Thanks for these thought-provoking comments. We agree with the reviewer that our study focuses on temperature-related death, rather than disease incidence or mortality rate. We recognize the value of disease incidence or mortality rate in measuring disease burden, but the two indicators cannot be calculated from the present death registry. However, virtually, **as the most stable and reliable health endpoint, death from governmental registry was widely employed in environmental epidemiological studies and assessment of burden of disease**¹⁻⁶.

We acknowledge the limitation that our study's scope is restricted to the deceased population. While death data provides important insights into the overall burden of diseases, **it does not capture the full spectrum of disease burdens**. We address this in the Limitations section of our paper, clarifying that **our findings provide specific insights into temperature-related death rather than representing the entire population affected by neurodegenerative diseases**. Please refer to lines 320-323.

We also made clear the interpretations of our results throughout the manuscript. For example, we stated "neurodegenerative diseases death risk" rather than "neurodegenerative diseases mortality risk", because we did not directly evaluate the

risk of neurodegenerative diseases in the population. We thereby replaced the term “mortality” with “death” in most places of the manuscript, including in the title.

References:

1. Liu C, Chen R, Sera F, et al. Ambient Particulate Air Pollution and Daily Mortality in 652 Cities. *N Engl J Med*. 2019;381(8):705-715.
2. Gasparrini A, Guo Y, Hashizume M, et al. Mortality risk attributable to high and low ambient temperature: a multicountry observational study. *Lancet*. 2015;386(9991):369-375.
3. Chen R, Yin P, Wang L, et al. Association between ambient temperature and mortality risk and burden: time series study in 272 main Chinese cities. *BMJ*. 2018;363:k4306.
4. Chen J, Gao Y, Jiang Y, et al. Low ambient temperature and temperature drop between neighbouring days and acute aortic dissection: a case-crossover study. *Eur Heart J*. 2022;43(3):228-235.
5. Zhou Y, Gao Y, Yin P, et al. Assessing the Burden of Suicide Death Associated With Nonoptimum Temperature in a Changing Climate. *JAMA Psychiatry*. 2023;80(5):488-497.
6. Jiang Y, Hu J, Peng L, et al. Non-optimum temperature increases risk and burden of acute myocardial infarction onset: A nationwide case-crossover study at hourly level in 324 Chinese cities. *EClinicalMedicine*. 2022;50:101501.

2. I am struggling making sense of the lag-response functions shown in Figure 2B. Most of them present RRs below 1 at lags 0-1 and some of them important RRs at the end. This makes me think that the lag dimension of the DLNM is perhaps not optimally set. It seems to me that the specification is roughly the one usually used in all-cause mortality, but perhaps in this context this is not the best one. Have any kind of model selection been performed by e.g. comparing the AIC or by performing sensitivity analyses to the lag-response curve?

Reply: Thank you for the comment regarding Figure 2B in our study. We acknowledge the complexity of the lag-response functions and appreciate your effort to understand them.

The parameter selection for the lag dimension of the DLNM was determined based on previous literatures on temperature-mortality relationships, typically ranging from days to weeks¹⁻⁴. We acknowledge that the lag-response functions depicted in Figure 2B may initially appear counterintuitive, with some relative risks (RRs) falling below 1 at lags 0-1. Nevertheless, this pattern aligns with many prior studies on the relationship between low temperature and mortality, **where this seemingly anomalous pattern has not been reasonably explained or had been overlooked**⁵⁻⁹.

To further investigate this issue, we reduced the lag length and specifically fitted the lag pattern for low temperatures. For a maximum lag of 3, we found that the risks at lags 0-1 became **insignificant or was changed to be positive and significant** (Figure R-1), but, the risks did not disappear at lag 3 d. When we extended the maximum lag to 14 d, the risk disappeared but the negative risks occurred in the first two lag days. This phenomenon is tricky, and varied by the selection of the maximum lag. We just reported these negative risks in the results section (see lines 130-133).

Following the reviewer's suggestion, we also compared various DLNMs using the Akaike Information Criterion (AIC), which gauges the balance between goodness-of-fit and model complexity. Our findings revealed that **the lag dimension (0-14 d) employed in the primary analysis provided the optimal fit for most models**. Please refer to lines 415-416.

Ultimately, in order to ensure comparability and consistency of results across regions and outcomes, we selected the uniform lag dimension for the main analysis. We highlighted this point in lines 416-418 in the method section.

Figure R-1. Lag-response curves for the relative risks of neurodegenerative disease death comparing extreme low temperatures (the 2.5th percentile) to the minimum-death temperatures, at the national and regional levels. The associations were presented as the cumulative relative risks comparing a given temperature to the minimum-death temperatures over lag 0–3 day.

References:

1. Gasparrini A, Armstrong B, Kenward MG. Distributed lag non-linear models. *Stat Med.* 2010;29(21):2224-2234.
2. Gasparrini A. Distributed Lag Linear and Non-Linear Models in R: The Package dlrm. *J Stat Softw.* 2011;43(8):1-20.

3. Chen R, Wang C, Meng X, et al. Both low and high temperature may increase the risk of stroke mortality[J]. *Neurology*, 2013, 81(12): 1064-1070.
4. Zhang S, Yang Y, Xie X H, et al. The effect of temperature on cause-specific mental disorders in three subtropical cities: a case-crossover study in China[J]. *Environment international*, 2020, 143: 105938.
5. Analitis A, Katsouyanni K, Biggeri A, et al. Effects of cold weather on mortality: results from 15 European cities within the PHEWE project. *Am J Epidemiol*. 2008;168(12):1397-1408.
6. Guo Y, Barnett AG, Pan X, Yu W, Tong S. The impact of temperature on mortality in Tianjin, China: a case-crossover design with a distributed lag nonlinear model. *Environ Health Perspect*. 2011;119(12):1719-1725.
7. Chen R, Yin P, Wang L, et al. Association between ambient temperature and mortality risk and burden: time series study in 272 main Chinese cities. *BMJ*. 2018;363:k4306.
8. Pascal M, Wagner V, Corso M, Laaidi K, Ung A, Beaudou P. Heat and cold related-mortality in 18 French cities. *Environ Int*. 2018;121(Pt 1):189-198.
9. Breitner S, Wolf K, Peters A, Schneider A. Short-term effects of air temperature on cause-specific cardiovascular mortality in Bavaria, Germany. *Heart*. 2014;100(16):1272-1280.

3. I would be more cautious in claiming a causal association simply because a case-crossover design has been used here. My understanding is probably incomplete but I think case-crossover and time-series studies are roughly equivalent in the environmental epidemiology context (see <https://doi.org/10.1093/biostatistics/kxl013>). I think one way to see this is that the individual confounders that are controlled for by design are not really confounders in time series designs anyway, since we don't expect them to vary across time, or they vary they will mostly be captured by time-related terms in the models. Additional information on causality in the case-crossover design can be found in this nice paper: <https://doi.org/10.1111/biom.13749>. Finally, note that the paper cited by the authors when claiming causality ([33] l. 307) does not really talk about that but is just another application of this design, and thus irrelevant here.

Reply: We appreciate these insightful advices and valuable references.

Firstly, we acknowledge the need for prudence when claiming a causal association based on the case-crossover design used in our study. We apologize for any confusion caused by the inadequate citation in our paper, which was removed from the revised manuscript.

Regarding the case-crossover design, it is essential to highlight that there are two distinct types: **the traditional aggregate-level design and the recent individual-level design**. The literature (<https://doi.org/10.1093/biostatistics/kxl013>) primarily focuses on the traditional case-crossover design, which showed equivalence between case crossover studies and time-series studies. In the present individual-level case-crossover study, exposure levels are derived from satellite measurements matched to residential addresses and the data were analyzed at the individual level rather than the aggregate level as done in earlier case-crossover studies. Our study design could reduce exposure measurement errors and well control individual-level, time-invariant confounders. **Nevertheless, due to the inherent limitations of observational studies, especially some unmeasured residual confounding (e.g., time-varying confounders), causal interpretations should be made with caution.**

We have revised the manuscript to avoid language indicating any causality and cited the relevant literatures accordingly. Please refer to lines 37, 230-232, 326-328, 344.

4. We don't get a good sense of the public health burden associated with temperature-related neurodegenerative diseases as no measure of prevalence is given. i think it would be interesting to either report attributable numbers, or death rates somewhere. No need to revolutionise the paper for this, but a few figures in the text would be relevant.

Reply: Thanks for this suggestion. We agree that this is an important consideration and have included additional information in the revised manuscript to address this point.

Firstly, we presented the **average of temperature-attributable number of neurodegenerative disease deaths** in each decade between 1980 and 2009. Please

refer to the below text or lines 168-172 and Supplementary Table 4.

“The average number of excess deaths from overall neurodegenerative diseases per decade between 1980 and 2009 attributable to cold and heat were 99,560 (95% empirical confidence intervals (eCIs): 83,032 – 115,909), and 9,999 (95%eCI: 7,511 – 13,226), respectively (Supplementary Table 4).”

We further showed **the differences in temperature-attributable numbers of neurodegenerative diseases deaths in 2010-2090 compared with 1980–2009 under three climate change scenarios**. Please refer to lines 185-192, lines 198-200, lines 464-466, and Supplementary Tables 7-8, Supplementary Tables 10.

“In the 2090s, the number of heat-related deaths of overall neurodegenerative diseases will increase by 15,384 (95%eCI: 6,455 – 26,185) under SSP126, 26,371 (95%eCI: 13,744 – 39,683) under SSP245, and 69,946 (95%eCI: 31,899 – 114,617) under SSP585, while the number of cold-related deaths of overall neurodegenerative diseases will decrease by 15,734 (95%eCI: 21,428 – 9,749) under SSP126, 22,569 (95%eCI: 33,774 –15,517) under SSP245, and 41,285 (95%eCI: 62,950 –28,147) under SSP585 (Supplementary Table 7 and Supplementary Table 8). In terms of the net changes in attributable number, there would be an increase of 3,569 in the 2050s and 28,715 in the 2090s nationally under the high-emission scenario (Supplementary Table 10).”

These results provide a more comprehensive understanding of the impact of temperature-related neurodegenerative diseases on public health.

Minor

l. 97: 0.43 million, so 430,000? Please use the same formatting as the following figures.

Reply: Thanks for pointing this out. We revised it to 430,000 to keep the same formatting. Please refer to line 95.

l. 100: does it represent a disproportionate amount given the population of each climate zone? Also, if other zone account for only 4% (each?), where are the remaining ~25%?

Reply: Yes, there is significant variation in the number of deaths from degenerative neurological diseases across different climate zones. This variation can be attributed to **the differences in population sizes** among the five climatic zones. Throughout the study period, the temperate monsoon, subtropical monsoon, temperate continental, tropical monsoon, and highland alpine climate zones accounted for **37.0%, 55.0%, 5.0%, 1.2%, and 1.8% of the national population**, respectively. Regarding deaths from degenerative neurological diseases, these climatic zones accounted for **33.4%, 62.6%, 2.9%, 0.5%, and 0.6% of overall neurodegenerative disease deaths**, respectively. For specific figures regarding the number of deaths in each climate zone, please refer to lines 100-103 and Supplementary Table 1.

Not sure both Figure 3 and 4 are needed as they roughly provide the same information.

Reply: Thanks for pointing this out. Actually, the two figures had different purposes. Figure 3 depicts the heat- or cold- related AFs for the future period, whereas Figure 4 presents the **difference** in AFs between the future and historical periods, including the **net change in trend**.

Higher risks in females can be explained by different age structure. Does the result hold when it is stratified by both age and sex?

Reply: Good point. We agree that the elevated risks observed in females may stem from variations in age composition. To address this, we added a new analysis stratifying combinations of age and sex. The findings indicate that **older females are more vulnerable** to the risks of death from neurodegenerative disease related to nonoptimum temperatures than older males. Please refer to lines 149-151, lines 439-441, and Supplementary Fig. 3.

1. 366 Truncating the extremes of temperature is a bit odd as these extremes could also have important effects. A sensitivity analysis to this would be interesting.

Reply: Thanks for raising this question. We acknowledge the reviewer's concern regarding the exclusion of temperature extremes in our analysis. It is worth noting that

throughout the analysis, we **incorporated all available data for modeling analysis** to establish exposure-response relationships for the complete temperature series, which were then utilized in the prediction study. When plotting the exposure-response relationship, we **limited our display to the 1st to 99th percentiles** of the temperature series. Importantly, this selection does not affect our study's main findings. We highlighted this point in lines 425-430 in the method section.

Truncating extreme values is a common practice when plotting exposure-response relationships¹⁻². **This helps mitigate the high statistical uncertainty and wide confidence intervals at exposure extremes due to the very small sample size.** To address the reviewer's concern, we performed a sensitivity analysis that **included temperature extremes.** Figure R-2 confirms the inverse J-shaped relationship between temperature and the risk of death from neurodegenerative diseases. Additionally, we found that the **relative risks were greater at temperature extremes, with significantly wider confidence intervals.**

Figure R-2. Exposure-response curves for the associations between daily mean temperature and death of overall and specific neurodegenerative diseases, at the national and regional levels. The

associations were presented as the cumulative relative risks comparing a given temperature to the minimum-death temperatures over lag 0–14 day.

References:

1. Chen J, Gao Y, Jiang Y, et al. Low ambient temperature and temperature drop between neighbouring days and acute aortic dissection: a case-crossover study. *Eur Heart J.* 2022;43(3):228-235.
2. Zhou Y, Gao Y, Yin P, et al. Assessing the Burden of Suicide Death Associated With Nonoptimum Temperature in a Changing Climate. *JAMA Psychiatry.* 2023;80(5):488-497.

Methods: I guess T_{mm} represents the MMT, but it should be defined more explicitly for less familiar audiences.

Reply: Yes, T_{mm} denotes the Minimum Mortality Temperature (MMT), which is the temperature that corresponds to the lowest mortality risk. As suggested, we clarified T_{mm} in the methods section (see lines 434-435).

Reviewer #3:

This paper on temperature-related mortality from neurodegenerative diseases adds important epidemiological evidence of climate impacts on health. The statistical power with large number of cases and the case-crossover design minimizing potential confounding are key advantages.

The inverse J-curve is not surprising and mirrors the relationship with all-cause mortality. Also, consistent with other temperature mortality studies, warmer climates have higher Min Mortality temp (MMT) suggesting some physiological or behavioral adaptation.

Reply: We appreciate the reviewer's positive and encouraging feedback.

As the reviewer noted, the most significant novelty of our research is the use of the **China's most representative death registry**. Furthermore, our study extends projections of future neurodegenerative disease death burden utilizing daily temperature simulations, which is **a novel approach**. Finally, we employed an

individual-level case-crossover study in a large-scale database, which **minimizes potential confounding factors** when compared to prior ecological time-series or case-crossover studies.

I think the authors overstate a causal relationship, mostly pointed to their case cross-over design (which is a strong design). I would like the authors to go further in explaining the robustness of the data going into this study, for example:

Reply: We acknowledge the need for prudence when claiming a causal association based on the case-crossover design used in our study. Regarding the case-crossover design, it is essential to highlight that there are two distinct types: **the traditional group-level design and the newer individual-level design**. In the present individual-level case-crossover study, exposure levels are derived from satellite measurements matched to each residential addresses, which could reduce exposure measurement error and well control for individual-level time-invariant confounders. **Nevertheless, due to the inherent limitations of observational studies, especially the unmeasured residual confounding, causal interpretations should be made with caution.**

We have revised the manuscript to avoid language indicating any causality. Please refer to lines 37, 230-232, 326-328, 344.

1) How reliable is death certificate data in China? Can they verify that the main cause of death is neurodegenerative disease for these death certificates?

Reply: Good point. We acknowledge the importance of accurate death certificate data in ensuring the validity of our study. The death data used in this study came from the China Cause of Death Reporting System (CDRS), **a well-established system designed by the central government** to collect information from all deaths occurring in China Mainland. The data collection process adhered to **stringent protocols, standard procedures, and meticulous quality control measures**. **Regular training and supervision** were carried out at all administrative levels, from township to national, throughout the year to assure the quality of each death certificate reported. **Experienced staff** in each district- or county-level Center of Disease Control and Prevention assigned

and coded the underlying cause of death using the International Classification of Diseases (ICD) coding system. As a routine quality control measure, **a random sample of death certificates** was selected monthly by the higher-level CDC to verify the underlying cause of death. The data from this system are widely used by the central government to produce official mortality statistics for informing health policy, as well as being a reliable data source for scientific research. Detailed descriptions of the registry were published elsewhere¹. We have provided more detailed information about the registry in the Methods section (see lines 355-366).

However, we acknowledge that **diagnostic errors may occur despite rigorous data verification and audit procedures due to the complexity and heterogeneity of neurodegenerative diseases**. We have included this limitation in text (see below text or lines 323-325).

“Sixth, despite rigorous data verification and audit procedures, diagnostic errors may occur due to the complexity and heterogeneity of neurodegenerative diseases.”

References:

1. Liu S, Wu X, Lopez AD, et al. An integrated national mortality surveillance system for death registration and mortality surveillance, China. Bull World Health Organ. 2016;94(1):46-57.

2) Related to (1) I think to be more credible, for the same locations (national and the 2 regions) we should see the all-cause mortality graphs. How different are they? Clearly these diseases are covariate with age (no surprise that women show higher response).

Reply: Thanks for pointing this out. As suggested, we additionally provide a table outlining statistics for all-cause deaths and total neurodegenerative disease deaths at both national and regional levels.

Table R-1. The statistics (proportion, %) at the national and regional levels for all-cause deaths and total neurodegenerative disease deaths, stratified by age and sex.

Region	All	Male				Female			
		Total	-64	65-74	75-	Total	-64	65-74	75-
All-cause death									
Nationwide	100.0	58.4	19.9	13.4	25.0	41.6	9.1	7.8	24.6
Subtropical monsoon zone	53.3	31.2	10.3	7.1	13.8	22.1	4.7	3.9	13.5
Temperate monsoon zone	41.0	23.8	8.3	5.6	9.9	17.3	3.8	3.5	10.0
Neurodegenerative disease death									
Nationwide	100.0	46.5	4.7	7.8	34.0	53.5	3.1	6.0	44.4
Subtropical monsoon zone	62.6	28.0	2.6	4.5	20.9	34.0	1.7	3.6	28.7
Temperate monsoon zone	33.4	16.6	1.9	2.9	11.8	17.7	1.2	2.1	14.4

We observed a higher proportion of total neurodegenerative disease deaths in the subtropical monsoon zone than in the temperate monsoon zone, which is comparable to the proportion of all-cause deaths in the two zones. Females accounted for 53.5% of total neurodegenerative disease deaths and the elderly (≥ 75 years) contributed to 78.4%.

3) A stronger discussion on potential physiological explanation would be useful. True that oxidative stress can always be a factor for many diseases. The authors in the discussion could offer more information, for example, on the near opposite relationships to temperature between Alzheimer's and Parkinson diseases.

Reply: We appreciate the valuable suggestion to include a stronger discussion on potential physiological explanations for the observed associations.

In the revised manuscript, we provide a **more comprehensive overview** of the potential mechanisms underlying the association between temperature and neurodegenerative diseases. Specifically, we discuss the role of **oxidative stress, inflammation, and changes in blood flow and metabolism in the brain** (see below text or lines 261-266).

“Further, temperature alterations may influence cerebral blood flow and metabolism, which are crucial for maintaining brain health. High temperatures may result in vasodilation and increase blood flow, leading to cerebral hyperperfusion. In contrast, low temperatures may induce vasoconstriction and reduce blood flow, potentially impairing cerebral metabolism and exacerbating neurodegenerative processes.”

It is worth noting that the temperature relationship between Alzheimer's disease and Parkinson's disease is not completely opposite, except that the effect of temperature may vary slightly depending on the region and the specific disease (see lines 137-138). Additionally, we included a discussion the effect of temperature on Alzheimer's disease and Parkinson's disease (see below text or lines 266-278).

“However, the effects of temperature could vary across different diseases due to their distinct pathophysiological characteristics. Notably, both high and low temperatures are associated with increased risks of Alzheimer's disease, albeit with potentially different underlying mechanisms. High temperatures are thought to promote amyloid- β aggregation and induce neuronal damage. Conversely, low temperatures can impair cognitive function and worsen pathological changes linked to Alzheimer's disease by reducing cerebral blood flow and metabolism. Regarding Parkinson's disease, the impact of high temperatures is more pronounced, while there is a weaker association with low temperatures, likely due to the exacerbation of oxidative stress and inflammation. Nevertheless, due to the complexities and intricacies involved in understanding the specific temperature effects on different neurodegenerative diseases, further research is necessary to delve deeper into these relationships.”

Reviewers' Comments:

Reviewer #1:

Remarks to the Author:

The authors have not taken into account the reviewer's indications on the need to calculate the heatwave definition temperature for each location and not to use a fixed percentile.

As already indicated in the previous review, the WHO itself states that: "Heatwave definition temperatures should be determined on the basis of epidemiological studies for each location and not on the basis of fixed meteorological percentiles".

<https://apps.who.int/iris/handle/10665/339462>

Therefore, by using a fixed percentile, the authors go against the WHO recommendations for this type of study. For this reason the manuscript cannot be accepted for publication.

Reviewer #2:

Remarks to the Author:

The authors addressed all raised points satisfactorily. I don't have any further comment.

Reviewer #3:

Remarks to the Author:

The authors have done a thorough job responding to reviewer comments and revisions to the manuscript.

Dear Editors and Reviewers,

Ref.: Ms. NCOMMS-23-10912A

Heartfelt appreciation to the dedicated editors and reviewers who played a crucial role in refining this article. Your constructive feedback and attention to detail have made a significant impact. We have revised the manuscript accordingly and provided point-to-point replies to these comments.

Editorial Requirements:

Reply: All the editorial requirements have been addressed accordingly. The authors declare that there is no conflict of interest.

Reviewer Comments

Reviewer #1 (Remarks to the Author):

The authors have not taken into account the reviewer's indications on the need to calculate the heatwave definition temperature for each location and not to use a fixed percentile.

As already indicated in the previous review, the WHO itself states that: "Heatwave definition temperatures should be determined on the basis of epidemiological studies for each location and not on the basis of fixed meteorological percentiles".

<https://apps.who.int/iris/handle/10665/339462>

Therefore, by using a fixed percentile, the authors go against the WHO recommendations for this type of study. For this reason, the manuscript cannot be accepted for publication.

Reply: We extend our gratitude to the reviewer for the insightful comments and concerns regarding the methodology for defining heatwaves. It is important to clarify that **our study did not explicitly define a heatwave but instead utilized extreme temperatures at the 2.5th and 97.5th quartiles to calculate the risk of neurodegenerative disease-related deaths relative to minimum-mortality temperatures.**

As the reviewer aptly pointed out, the World Health Organization has emphasized the relative nature of heatwaves concerning a location's climate. Defining temperature extremes using **an absolute threshold** across different locations is inappropriate. In the realm of epidemiology, studies using **fixed/uniform relative percentiles** for defining temperature extremes have undergone rigorous peer review, gaining recognition as **valid methods** for assessing the health impacts of temperature extremes nationally or regionally^[1-7]. **This approach ensures consistent assessment across diverse locations and time periods.** Our approach aligns with this methodology, employing a uniform relative threshold to define temperature extremes, **consistent with prior high quality literature in reputable journals** such as The Lancet^[1], The British Medical Journal^[2-3], and European Heart Journal^[4]. Importantly, to the best of our knowledge, there are no existing studies defining temperature extremes using temperature thresholds directly determined from epidemiological studies for each location.

Crucially, **the definition of temperature extremes only influences risk estimates at a given temperature and does not impact the shape of exposure-response curves or the assessment of disease burden due to overall nonoptimal temperatures.** This is because our analysis incorporated all available data for modeling, establishing exposure-response relationships for the complete temperature series, which were subsequently utilized in prediction studies.

References:

1. Gasparrini A, Guo Y, Hashizume M, et al. Mortality risk attributable to high and low ambient temperature: a multicountry observational study. *Lancet*. 2015;386(9991):369-375.
2. Chen R, Yin P, Wang L, et al. Association between ambient temperature and mortality risk and burden: time series study in 272 main Chinese cities. *BMJ*. 2018;363:k4306.
3. Sun S, Weinberger KR, Nori-Sarma A, et al. Ambient heat and risks of emergency department visits among adults in the United States: time stratified case crossover study. *BMJ*. 2021;375:e065653. Published 2021 Nov 24.

4. Chen J, Gao Y, Jiang Y, et al. Low ambient temperature and temperature drop between neighbouring days and acute aortic dissection: a case-crossover study. *Eur Heart J*. 2022;43(3):228-235.
5. Masselot P, Mistry M, Vanoli J, et al. Excess mortality attributed to heat and cold: a health impact assessment study in 854 cities in Europe. *Lancet Planet Health*. 2023;7(4):e271-e281.
6. Zhou Y, Gao Y, Yin P, et al. Assessing the Burden of Suicide Death Associated With Nonoptimum Temperature in a Changing Climate. *JAMA Psychiatry*. 2023;80(5):488-497.
7. Jiang Y, Hu J, Peng L, et al. Non-optimum temperature increases risk and burden of acute myocardial infarction onset: A nationwide case-crossover study at hourly level in 324 Chinese cities. *EClinicalMedicine*. 2022;50:101501.

Reviewer #2 (Remarks to the Author):

The authors addressed all raised points satisfactorily. I don't have any further comment.

Reply: Thank you for your thorough review and positive feedback. We appreciate your time and are pleased to hear that the addressed points were satisfactory. Your insights have been invaluable to the improvement of our work. Thank you once again for your time and consideration.

Reviewer #3 (Remarks to the Author):

The authors have done a thorough job responding to reviewer comments and revisions to the manuscript.

Reply: Thank you for your thoughtful review and acknowledgment of our efforts in addressing reviewer comments and revising the manuscript. We appreciate your positive feedback and are pleased to know that our responses have been thorough. Your insights are invaluable, and we are grateful for your time and expertise in reviewing our work.